# Social Responsibility, Sustainability, and Public Policy: The Lessons of Debris Management after the Manabí Earthquake in Ecuador

**DOI:** 10.3390/ijerph18073494

**Published:** 2021-03-27

**Authors:** Paulina Guerrero-Miranda, Arturo Luque González

**Affiliations:** 1Facultad de Ciencias Agrícolas, Universidad Central del Ecuador, Ciudadela Universitaria, Av. América y Avda. Universitaria, Quito 170136, Ecuador; jpguerrerom@uce.edu.ec; 2Departamento de Análisis Geográfico Regional y Geografía Física, Universidad de Alicante, Carretera San Vicente del Raspeig s/n, San Vicente del Raspeig, 03690 Alicante, Spain; 3Facultad de Ciencias Humanísticas y Sociales, Universidad Técnica de Manabí UTM, Ave. José María Urbina and Che Guevara, Portoviejo 130105, Ecuador; 4Escuela de Administración—Grupo de Investigación en Dirección y Gerencia, Universidad del Rosario, Bogotá 111711, Colombia

**Keywords:** earthquake, recycling, debris, waste, sustainability, land regulation, Ecuador

## Abstract

Natural disasters can generate millions of tons of debris and waste, which has an impact on the environment and poses direct risks to the health of the population, hence the need to analyze public policy and its consequences following the 2016 earthquake in Ecuador. Several in-depth interviews were conducted with individuals active in public service during the post-earthquake management period, together with fieldwork analysis of debris management and the institutional strategies for its recycling and reuse in three of the most affected cities: Pedernales, Portoviejo, and Manta. The environmental impact was examined, including its taxonomy of inconsistencies within public administration, alongside the processes of decentralization and shared decision-making. Similarly, the links between corporate social responsibility (CSR), public policy, and sustainability were analyzed at both the national and local level for their wider implications and ramifications. The study highlighted the gaps in the management of such a crisis, exposing a lack of ethics and the shortcomings of social (ir-)responsibility in the distorted processes of public welfare in the country, aspects that should rather work in concert to achieve full sustainable development.

## 1. Introduction

On 16 April 2016, Ecuador suffered one of the most devastating tragedies in its history, an earthquake of magnitude 7.8 that affected the coastal provinces of Manabí and Esmeraldas. The disaster left 663 dead as well as a large number of missing persons whose whereabouts are still unknown today. In the following months, more than 2300 aftershocks were recorded, several larger than 6.0 in magnitude. In total, 9901 jobs with a combined income of $16 million were lost, and additional costs of $18 million accrued from the impact on agriculture, manufacturing, trade, food and drink services, accommodation, and other sectors. The earthquake destroyed nearly 7000 buildings and a further 1000 were severely affected, which gives some idea of the scale of the disaster and the proceeding earthquake management [1]. The final total of losses estimate was at $354 million [2].

In addition to this, the earthquake generated a vast quantity of waste material. This construction and demolition waste (C&DW) is a diverse collection of materials commonly referred to as “debris” and is commonly one of the most significant consequences of the destructive power of earthquakes due to the large number of collapsed buildings. After the earthquake, the management of debris from damaged urban infrastructure was one of the priority tasks to be carried out, not only due to its immediate impact on the search for survivors, but because of the need to re-open roads and public spaces in order to start the rebuilding process for the benefit of the affected populations. The social upheaval and chaos caused by events of this kind rightly leads to the prioritization of human lives by the authorities, together with the need to make urgent decisions on a wide variety of other issues, many of which do not prove to have been the most appropriate. All of this is imbued with the need for rapid and far-reaching aid and solidarity [3].

If, under normal conditions, solid household waste management is often deficient and in need of improvement, it should not be surprising to find significant problems in dealing with waste material in the event of a disaster and public emergency; these issues are the subject of this study. Furthermore, this study focused on three aspects: (i) Debris management that was analyzed in the three cities most affected by the April 2016 earthquake: Pedernales, Manta, and Portoviejo; (ii) interviews and other fieldwork conducted with technical staff working for local authorities; and (iii) a five-process model that was applied based on the assessment of the affected infrastructure in each city and the attendant issues of debris removal to disposal sites. The overarching objective was to understand the weaknesses and opportunities arising from such a catastrophe.

## 2. Debris

Whether the product of normal demolition processes or of the effects of an earthquake, debris can be reused as a raw material for construction purposes, making its analysis essential to discover what proportion may be recycled and what treatment processes need to be used. At the same time, adequate arrangements must be made for the disposal of the remaining debris. As a reference, studies carried out on debris in Bogota [4], Barranquilla [5], and Madrid [6] were reviewed. In the latter case, at the Valdemingómez Technological Park, analysis revealed that 78% of urban waste is subject to some treatment, with the remaining 22% being sent directly to landfill [7]. The waste material contained in these studies was not the product of earthquakes but of normal demolitions, yet it is significant to note that, as a general rule, some 80% of materials can be recycled and reused in new constructions. Knowing the proportion of debris that can be recycled and reused and that a relatively small amount needs to go to the disposal sites makes it clear that the management of this waste requires a careful approach.

After a catastrophe, waste recycling processes depend on many factors, such as: (1) the amount generated, (2) precise mixture of the waste materials, (3) range of waste materials, (4) priorities of the affected community and its interrelationships, (5) endogenous and exogenous mechanisms of finance, (6) recycling market, (7) dangers to human and environmental health, (8) time constraints, (9) available technology, and (10) existing legislation on both general and specific disasters [8]. In the midst of a crisis, debris represents an opportunity to have ready raw materials for the city’s reconstruction. In this sense, it is imperative that interventions by local, provincial, and national authorities are based on more effective public management tools that take an integrated approach to cities, especially since mismanagement affects the population through health and environmental risks. Additionally, and not least in the case of coastal Ecuador, there is the consideration of the impact on tourism, as poor waste management makes for less attractive destinations [9].

### 2.1. Post-Disaster Debris in Other Countries

In the literature on post-disaster reconstruction, very little is documented of the management of debris, and a limited number of countries that have experienced earthquakes have succeeded in reusing these materials, generally through local initiatives with the help of international organizations, such as the United Nations Development Programme. The UNDP, as part of its humanitarian action in international cooperation, implements “emergency employment projects” aimed at increasing the purchasing power and consumption of those affected by enabling them to meet their immediate needs and reduce vulnerability while recovering from crises. One of these projects is the “Cash for Work” modality that involves the provision of wage payments in exchange for labor through various short-term activities. In addition to the removal, recycling, and reuse of debris in cities affected by conflicts, UNDP has supported post-earthquake actions in various locations [10], such as (1) Indonesia, with the cleaning up more than 1,000,000 m^3^ of debris caused by the 2004 earthquake and tsunamis; (2) Pakistan, with the removal of 554,030 m^3^ of debris following the 2005 earthquake; (3) Haiti, with the disposal of 10,000,000 m^3^ of debris in the 2010 earthquake; and (4) Italy, in many episodes, including Sicily (1968), Friuli-Venecia Giulia (1976), Campania (1980), Umbria and Marche (1997), Molise (2002), Abruzzo (2009), Emilia (2012), Umbria, Lazio, and Marche (2016 and 2017) [11] (Shirvani et al., 2019).

Among the experiences of post-earthquake management through the partnership of private enterprise together with local authority initiatives is that of the city of Los Angeles (USA) after the 6.7-magnitude earthquake in 1994. The amount of debris generated was estimated at 3,000,000 tons by the end of July 1995. This city developed one of the enduring models of debris collection in a disaster zone by using private collaboration and the transportation of debris by rail. Another case was Mexico, where the Department of Environment and Natural Resources (DENR) issued the “criteria for the management of construction and demolition waste generated by the September 19 earthquake for the states of Mexico, Morelos, Puebla, and Mexico City”. This document provides technical guidelines for the location and operation of sites for the final waste disposal and the establishment of temporary warehouses, but, above all, it promotes the use and recycling of debris [12]. After the earthquake in Chile on 16 September 2015, the nonprofit organization Biobio Proyecta was created in the city of Concepción. Originally, it was oriented only toward dealing with the debris of large buildings and it established fundamental approaches for the management of this type of waste. The Memory Project proposed by this NGO took “identity” as an aspect linked to the infrastructures that, when lost or demolished, risked making Chile appear as a country that does not safeguard its heritage. In response to this problem, they created a category of symbolic debris for those elements of value that must be salvaged because they are constructions that are part of the collective national memory according to Basoalto et al. [13].

All debris, without distinction, was sent to improvised dumps. In some cases, they were used for landfill, or in others they were simply stacked for recycling purposes. The important and urgent thing was to “get the waste out of town.” (...) We studied the issue, concluding that debris can possess value (collective, historical or aesthetic), if the connection is made with its place of origin and the idea of its being a potential element for the preservation of local memory. Although it is a fragment, it once belonged to an important building, it therefore becomes an attractive raw material for the construction of new heritage spaces”.

The last case concerns the government of Bangladesh, which, backed by UNDP [14,15,16]) and the Department of Disaster Management, issued guidelines for handling large volumes of post-disaster debris. It designed management programs that provide for initial assessments of the accumulation of this waste through to the actual implementation and end-use of recycled materials. The programs also promote sustainable livelihoods for the people involved in this process [17].

### 2.2. Post-Disaster Debris in Ecuador

The low recurrence of earthquakes means that such phenomena are given low priority in public policy and regulations. This is compounded by government agendas that are limited to managing the period of the incumbent’s administration. This is the case in several countries, including Ecuador, where debris from disasters has become part of the waste materials dealt with as part of demolition or construction activities. In fact, they are not considered in need of any separate treatment and, therefore, have generally been handled in the same way as any other waste using existing regulations for all solid waste. The Ecuadorian Ministry of Environment (EME) is the national environmental authority with powers for the definition and implementation of policies. In 2002, it carried out an “Analysis of Solid Waste in Ecuador by Sector” and, in 2010, the National Program for the Integrated Management of Solid Wastes (NPIMSW) was created. In May 2015, the EME issued Ministerial Agreement No. 61, which, among other issues, deals with three fundamental aspects: (a) mandating the comprehensive management of non-hazardous solid waste, and hazardous and/or special wastes; (b) establishing the responsibility of local authorities for the integrated management of waste generated in the local area, minimizing its accumulation, promoting its reuse, and giving it adequate treatment and final disposal; and (c) creating the need to promote economic usage programs, incorporating recycling and reuse processes [18] (EME, 2015).

In parallel to the environmental regulations, Article 136 of the Organic Law on Territorial Ordering mandates that local authorities have comprehensive waste management systems [19], enabling each authority to create its own regulations for comprehensive and responsible debris management. Consequently, there is, up to a point, a regulatory basis for the comprehensive handling of debris. However, despite this legal advantage, no authority has yet implemented specific technical and operational procedures for this type of management, at least in practice; most have merely complied with the three basic processes of waste management—collection, storage, and final disposal—and have prioritized domestic materials.

### 2.3. Brief Description of the Disaster

On 16 April 2016, at approximately 18 h 58 min, an earthquake of local magnitude (ML) 7.8 and an intensity of VIII occurred, with its epicenter between the parishes of Pedernales and Cojimíes, in Pedernales County, Manabí province, constituting the most destructive seismic event in the country since 1987 [20]. According to the map of Seismic Intensity by Parish (Figure 1), the provinces of Manabí and Esmeraldas were the most affected, with a total of 35 parishes recording intensities in the categories of “very high”, “high”, and “medium”.

Parishes classed as “very high” suffered severe damage to their infrastructure. Some were rural with a low population and housing density and, although this does not lessen in importance the degree of impact (including on society), the destruction was not comparable to the damage in the cities of Portoviejo, Manta, and Pedernales. In these cities, the size of the urban area along with its intrinsic characteristics made them more vulnerable to seismic events.

For example, Pedernales, in addition to being the epicenter of the earthquake, is an important tourist-beach destination for the central parts of the country; its hotel infrastructure, restaurants, and other tourist services were severely affected. Meanwhile, Portoviejo, the most populous city and capital of the province of Manabí, sustained extensive damage, as did Manta, the second most important seaport in the country, which saw the collapse of several buildings, including the control tower at Eloy Alfaro International Airport. In total, according to the Department of National Planning and Development [22], 80,000 people were displaced by the total loss of or significant damage to their homes, equivalent to 14% of the population in the three cities; similarly, the reconstruction cost here amounted to USD 3343.8 million, with the social sector being the most affected at 1368.6 million, or 40.9%. For housing and education infrastructure alone, the costs were $652.8 million and $434.8 million, respectively, equivalent to $47.7% and 31.77% of the sector.

## 3. Research Design

This research was based on a synthetic analytical methodology whose purpose was to deconstruct the problem into smaller parts for it to be optimally addressed [23]. These component parts are (1) a general review of national, provincial, and local legislation in Ecuador; (2) a review of similar processes and statistics in similar disasters; and (3) in-depth interviews conducted between July and August 2017 with experts involved in the reconstruction process, as well as active representatives of local authority institutions. It should be noted that the study encountered some difficulties in obtaining information from officials or technicians within local authorities, who sometimes felt the need to protect the institutional image if they thought that the information provided could be misunderstood by certain government authorities. The questions were designed based on related research in this field [9,24,25], and the interviews had four dimensions: impact/process assessment, effectiveness of national regulations, management practices, and suggestions for improving processes.

## 4. The Debris: A Problem to Be Solved

Ecuador, having not undergone seismic events of this magnitude in recent decades, did not have a point of reference for the management of debris, an area in which the intervening ministries are Transport and Public Works (MTPW), Urban Development and Housing (MUDH), and the Ministry of Security Coordination (MSC). Consequently, the “Protocol for the Demolition and Debris Removal Process” was hurriedly prepared, aimed at “providing an instrument to guide interventions in the demolition of damaged buildings and the removal of debris, considering the environmental management of debris, infrastructure assessment, legal standards and the principles and powers of each institution” [26]. The protocol was completed on 30 May 2016, some six weeks after the earthquake and was the main temporary measure to be adopted in answer to this problem. Prior to the validation of the protocol, some of the affected city authorities made use of their political, administrative, and financial autonomy in land management, as granted in Article 238 of the Constitution of the Republic [27], and carried out debris removal operations according to the criteria of local officials and the availability of machinery provided by private companies. The Ministry of the Environment was in charge of authorizing the selection of the final disposal sites. Other public institutions intervened in the area as follows: In Pedernales, the Civil Engineers Corps of the Army, in coordination with the Metropolitan Mobility and Public Works Company (MMPWC) of the Municipality of the Metropolitan District of Quito; in Portoviejo, City Hall of Portoviejo, together with MTPW, MUDH, and UNDP; and in Manta, City Hall of Manta through COSTALIMPIA, a company responsible for the collection and management of solid waste in this city. The specific actions undertaken in the cities of Pedernales, Portoviejo, and Manta were compiled from a model covering five chronological processes proposed according to the observations of the fieldwork (Table 1). These are: (a) Process 1: Problem assessment; (b) process 2: Initial activities in the management of debris; (c) process 3: Selection and authorization of final disposal sites; (d) process 4: Recycling of debris and other inherent activities; and (e) process 5: Final activities at final disposal sites.

Local authority autonomy in land management allowed for an immediate response to the emergency (the day after in all cases), even though they did not have the technical or operational capacity to deploy a comprehensive solution to the problem. The decentralized nature of the Ecuadorian state proved itself an important tool for local intervention despite the social and institutional chaos witnessed in the management of debris, which was manifest in the following aspects: (1) The response to the crisis was different in each city; (2) the most important processes for its management were subject to the criteria of the technicians of the local authority and the relevant ministries (in some cases, social and health institutions participated); (3) the evident solidarity shown by the rest of the country for the management of the crisis—even the Municipality of the Metropolitan District of Quito, despite its remoteness from the affected area, lent human resources and machinery for the 3 months following the disaster and private enterprise also actively contributed with machinery for each of the affected localities; and (4) in the absence of a protocol for the process of demolition and removal of debris, in the first 45 days after the earthquake, each authority carried out the work based on the knowledge of their own technicians, despite the existence of indicators and recommendations established at the global level. In fact, due to the large amount of debris generated by the earthquake, the selection of “final disposal sites” or “tips” proved challenging, and mistakes were made, particularly in Manta’s choice of “la Poza”. This was a recreational area and, therefore, inappropriate for this purpose, as was a local ravine that was considered a conservation area. The disposal operation at these sites was improvised and was not authorized by the EME, nor were the minimum requirements met to avoid negative impacts from pollution to the environment, urban landscape, and society. Hence, a catastrophe, such as an earthquake, tsunami, or tornado, may have unforeseen ramifications [28].

Job losses from the absence of infrastructure and places of work that had either been destroyed or damaged led to the emergence of a significant number of informal salvage workers. These individuals lacked technical procedures for the selection of materials for salvage but employed manual tools to break up debris and obtain ferrous or other recyclable materials for private sale. This work was incompatible with an effective recycling process. Faced with such a magnitude of debris (and all kinds of collateral waste), this salvage work succeeded in preventing only a minimum amount of recyclable material from disposal in the waste tips.

International cooperation through UNDP [29] (2020) allowed the delivery of safety equipment for salvage workers in the “la Solita” tip in Portoviejo. It also provided technical training to 594 volunteer architects and engineers to conduct assessments of buildings; however, it was not possible to implement this approach in the other cities. All three locations took advantage of some of the finer materials produced by the grinding down of debris in order to pave secondary roads in Manta and Portoviejo and in the hostel in Pedernales, proving how useful these waste products could be. Although the exact data on the amount of debris processed is unknown, the DNPD [22] estimated there was some 3251,150 m^3^ in Manabí Province, although this is an approximate figure as there are localities that, due to their remote location and the difficulty of reporting, are not included in the official statistics. On the other hand, determining the quantity or volume of a cubic meter of material involves weighing it, and that task proved difficult or almost impossible in the affected area owing to the magnitude of the destruction and the lack of adequate infrastructure and equipment to carry this out. Nonetheless, if from the outset a record of the amount of debris taken to the final disposal locations had been kept by quantifying the number of trips and the load capacity of the dumpsters, a thorough analysis of waste materials for effective recycling would have been possible.

### 4.1. The Imminent Environmental Impact

The EME authorized the operation of nine tips throughout the province of Manabí; of these, only the cities of Manta and Portoviejo had the necessary technical specifications for the use of machinery and other equipment for the final disposal of debris. However, having heavy machinery or having official approval is no guarantee that the deposited material will be less polluting than another that does not have approval. In order to understand the magnitude of the problem, it is necessary to take into account the space required to dispose of debris [30]. In the 2015 earthquake in Chile, the Urban Platform [31] noted: “If the National Stadium were a large waste tip, it would take almost three of them to contain one and a half million cubic meters of debris that is estimated to have been removed from the Metropolitan Region in the wake of the earthquake.” By the same token, the debris estimated to have been in Manabí alone would have needed more than six 48,000-seater stadiums for its disposal. This analogy perhaps gives a clearer idea of the scale of the problem and the vast space required for the disposal of these wastes. Further to the sheer quantities involved, and the fact that the makeshift waste tips were not adequately conditioned for receiving debris, the final disposal of debris that had not been properly treated for the separation of hazardous components generated pollution and led to various environmental impacts. Underscoring all of these observations is the Sendai Framework. “The Sendai Framework for Disaster Risk Reduction 2015–2030 outlines seven clear targets and four priorities for action to prevent new and reduce existing disaster risks: (i) Understanding disaster risk; (ii) strengthening disaster risk governance to manage disaster risk; (iii) investing in disaster reduction for resilience; and (iv) enhancing disaster preparedness for effective response, and to “Build Back Better” in recovery, rehabilitation and reconstruction” [32], “Building Back Better” sets out the essential requirements for any post-disaster reconstruction process, taking into account the interconnectedness of communities, as well as their resilience [33,34].

Table 2 shows the impact of the management and final disposal of debris on various aspects, including the direct impact on natural resources. It should be noted that these are not readily identifiable, so in order to quantify and qualify the real environmental impacts, local authorities—in fulfilling their decentralized responsibilities—have a duty to carry out the necessary environmental studies, as well as to perform physical-chemical analyses of the tips and their surrounding sites.

Environmental damage occurred in two phases: (1) in the processing of debris where pollution and impacts are related to collection and transport and (2) at final disposal sites, where waste remains permanently, generating several environmental liabilities. In both phases, the level of contamination is a function of the amount of material managed. The reduction of impacts does follow from having the relevant authorizations for the operation of a waste tip, let alone from transporting waste by secondary routes where fewer people might complain, since this only amounts to “transferring the site of the damage”. In the burning off of waste products, both legally and illegally, toxins are produced and emitted. These are chemical compounds with strong toxic, carcinogenic, and mutagenic effects that have been linked to immune suppression in humans [35,36]. Hence, there is a need to evaluate public policies differently, not only for their costs, but also for their consequences and to seek the improvement of legislation. In any event, unproductive marketing or social responsibility campaigns should be avoided, especially when these lack the means of verification and control and are not backed by true political will [37]. Overlooking resolutions and regulations or the failure to enact new legislation is tantamount to having a poor vision of social and environmental benefits and weak public management, especially if the earthquake is not seen as an economic opportunity in a time of crisis for a population that has lost everything.

### 4.2. Debris: The Result of a Crisis vs. Opportunity

For debris to be seen as an opportunity regardless of whether it comes from demolitions or the result of earthquakes, it can be supposed that it formed part of previous infrastructures and therefore demanded the mixing of certain materials, hence its composition as a building material does not change after the construction comes down. Consequently, it is essential that debris is taken advantage of as a resource, aided by introducing transformational and sustainable policies in the construction and recycling industry, based on the processes of a circular economy [38,39,40,41,42,43].

Various international experiences have shown that, even with international aid, only a small percentage of debris from earthquakes is recycled and reused for other purposes; the likely reason is the limited research in this area and the lack of knowledge of which materials might be exploited for different uses. For some managing authorities, debris is little more than garbage to be disposed of, further compromising the preservation of cultural heritage. In contrast, some cities have already carried out studies to determine not only the composition of potential debris, but the percentage of materials that may be recycled and reused, and which might not be recycled and what treatment should be employed for their disposal [44,45].

For buildings damaged by Hurricane Katrina in New Orleans, there was a total recovery rate of buildings of 48% (ranging from 28% to 62%) [46]. In Bogota, Colombia, a guide was developed for the comprehensive management of debris, placing them in three categories: “usable”, “non-usable”, and “others” (undefined). This research focuses on the “usable” category, i.e., those that can be reused, recycled, and revalued [47]. To determine the percentage of use in the list of usable materials in the Bogota guide, these were cross-referenced with studies regarding the composition of debris in Barranquilla, Colombia, and in Madrid, Spain.

The result is shown in Table 3.

The table shows that Madrid defined 9 reusable materials, of which 54% are bricks, tiles, and other ceramics, followed by concrete (12%) [48]. In Barranquilla, there are 10 usable materials with the highest percentage being aggregates (24.11%), followed by concrete and cement (18.16%). The difference in materials in both cities may be due to the construction system used in each and this is a difference that will also be found between other localities around the world; what is important is the total percentage of usable materials, which in the case of Madrid is 83.8% and in Barranquilla 88.53%. If Colombia and Ecuador have a similar construction system, the percentage of usable material should be around 88%. Therefore, with approximately 3,251,150 m^3^ of debris in Manabí alone, a figure of about 2,861,012 m^3^ of material could have been used as raw materials for new products. In addition, only 12% of the debris would need to have been buried at the final disposal sites, significantly reducing the environmental impact. Not taking advantage of a large quantity of debris misses a great opportunity in various areas. In the area of work, the U.S. Environment Agency made a comparison between the different types of management for this waste. In this analysis, one job would be created for every 10,000 tons of incinerated waste, while the operation of a tip could lead to 6 new jobs; however, recycling the same amount of waste would result in 36 jobs. From the economic perspective, the market for recycled building materials recorded revenues of 744.1 million euros in 2010, and by 2016, this had grown to an estimated 1.3 trillion euros [48]. Yet the economic advantages should not be seen only in terms of income, but also as saving. According to the Bangladesh Ministry of Disaster Management and Relief [17], some types of debris can be reused directly without any mechanical processing; indeed, with nothing more than its classification and some cleaning, for example, debris, such as bricks, stones and building blocks can often be immediately reused and incorporated into reconstruction work. If recycled products have the same degree of usefulness as new materials, it seems unnecessary to resort to expenses or investment in buying raw materials, especially if there is a sudden unexpected crisis, such as post-disaster reconstruction. The economic and employment advantages of debris recycling are easily observed, but what other uses can be given to “usable” materials, processed or unprocessed? As stated by the Department of the Environment and Natural Resources [12], several types of infrastructure works can be carried out with salvaged debris. This includes filling materials for engineering projects, gabions, blocks, sub-base for roads or parking lots, aggregate for concrete, scrap metal, embankments, landfills, roads and roofs, paths and cycle-paths, construction of beds for pipes, hydraulic systems or paving, and causeways, among others. In the vanguard of this type of waste management are countries, such as Germany, Spain, and Belgium, which have adopted a policy of separation at source and employ specific treatments and uses in different areas of construction, thereby, reducing the percentage of residual material to be disposed. Some countries have already used recycled debris in certain construction projects, one of which is Germany, where 56% of the recycled aggregates produced in 2008 were used as a base and sub-base in road construction and 30% for landscaping [48]. In the city of Los Angeles, with support from the Federal Emergency Management Agency and local businesses, three months after the debris removal process following the earthquake, it was decided to process and recycle as much as possible to conserve landfill capacity. Contracts were signed with companies who were provided with cleaned and separated materials, while a project was undertaken to recycle mixed debris. After one year, more than 10,000 tons of processed debris had been obtained [50]. Another example is Haiti, where in the 2010 earthquake, the International Labor Organization (ILO) trained potential entrepreneurs in recycling and debris processing. Their endeavors produced recycled materials, including 207,000 cobblestones with which more than 13 km of road were re-surfaced and many houses were rebuilt in the affected neighborhoods, “resulting in a cycle that promoted employment, accelerated infrastructure rehabilitation and boosted the local economy” [51].

While it is not a question of resorting to Keynesian theories of “crises” generating “opportunities”, debris has been shown to contain a large percentage of materials that can be recycled and reused and this brings with it a number of environmental, economic, and employment benefits. Re-using more than 80% of materials not only fosters environmental awareness in the population, but also prevents an overburdening of natural resources. It reduces the exploitation of mines and quarries and stops the use of natural spaces, such as ravines, from becoming polluted tips, whose clean-up and rehabilitation is costly and takes a great deal of time.

### 4.3. New Challenges for Local Authorities

What could reasonably have been expected of local authority debris management? The answer is not so simple, as proper management will depend to a large extent on the government’s commitment to sustainability processes, in this case, the harmony between economic, social, and environmental concerns. However, as one of the member states that adopted the 17 “Sustainable Development Goals” proposed by UNDP in 2015, and in seeking to fulfill the political commitment to sustainable urban development set out in the New Urban Agenda in 2016, Ecuador has an inescapable responsibility to think of new ways of implementing its land management policies, and local authorities are part of this process. While “in some discussions public sector institutions are negatively assessed as lethargic, fossilized bodies, and therefore unable to promote change” [52], local authorities are able to make use of their most valuable powers as established by the constitution: autonomy and decentralization. Through these, the changes required to implement development mechanisms are possible. In this context, two challenges arise for the general management of debris by local authorities: a new land management model is required, from the starting point of a spatial and functional organization of activities to the management of all of the resources of the areas they govern. This implies looking to the future, realizing that land is not immune from new crises and, if one should occur, the authority must be supported and have the full confidence of the population that the actions they undertake will help minimize the damage.

The model should consider coherent land management supported by planning, with provisions established for crisis care, environmental management, and a new approach to land uses. The latter, above all, is of great importance if contamination of natural spaces is to be prevented and the skills of the local population are to be respected. In the words of Gómez and Gómez [53], means, “putting all things in their place; the “things” to be organized are human activities, which will need to be identified while the “place” is the land; but using this organization requires regulating the way it is used. Therefore, organizing land means identifying, distributing, arranging and regulating human activities in that land area according to certain criteria and priorities.”

This concept forms the background to land management since putting everything in place means having regulations for land use, in which no human activities are permitted to violate the natural order, let alone diminish or affect its functioning. In the case of debris, regulations should contemplate the spaces potentially earmarked as disposal sites, and even regulate spaces, such as temporary warehouses, for classification. In both cases, ravines, beaches, or other natural areas are not to be used; rather, they should be the subject of conservation and protection.

Such regulation would prevent “improvised actions” and other blunders in favor of a new approach to debris management based on the “circular economy”, bearing in mind that “buildings are born, grow and die like living things. However, the ecological footprint of their passage through this world is far from ideal (...) debris often ends up in a tip or, at best, used to fill some hole in the ground” [54]. Considering this, the costs of recovery and reconstruction are much higher than those of prevention and mitigation, and applying the criteria of the circular economy goes beyond simple collection and final disposal. This occurs by incorporating important, such as promoting research and development of new technologies for the separation of materials and applying innovative techniques for the quantification and qualification of debris, including the recycling and reuse of materials as raw material for new constructions. Since an entire system exclusively for the management of debris from disasters cannot be expected, especially as such events are difficult to foresee, applying the principles of the circular economy to the processes of the dynamics of the city is the starting point for institutional improvement and its preparation to face crises of even greater proportions. The circular economy would help to galvanize the local financial situation by adding value to otherwise worthless debris and by promoting programs for its effective use. At the same time, this would foster the creation of formal employment at each stage of the comprehensive management process, allowing salvage workers to have a more dignified job less injurious to their health. In facing these challenges, local authorities help the country comply with SDGs and contribute significantly to the resilience of its cities.

## 5. Conclusions

The results show that the socio-economic crisis added to the institutional weakness of local authorities led to the management of debris being focused on collection and final disposal in improvised tips, despite the indications of public policy in the area of comprehensive waste management.

In the manner in which the crisis was addressed, post-earthquake debris management brought about hurried actions that reacted to the emerging situation. Despite the fact that Ecuador has been frequently subjected to natural threats, including seismic activity and earthquakes, little has been put into action to provide a preventive model of risk management and disaster reduction at all levels and in an effective manner. While there is regulation in place for comprehensive waste management, there are no specific public policies or regulations for debris management; having these instruments is critical to successful land management. A clear set of guidelines is indispensable, one that unifies management criteria, not only for earthquake debris, but also for waste from demolitions and the typical damage to structures that occurs in the course of urban dynamics.

Although each local authority handled the crisis differently, all were capable of having a more environmental and landscape-friendly, even humane, view of disaster management. It is also clear that the institutions that make up the state apparatus should consider a number of mechanisms to prevent not only such emergencies, but the ensuing waste of resources. Much of the capital available during the emergency phase was invested in demolishing damaged structures and discarding debris—the most abundant raw material—when resources could have been better allocated to investment in both small- and large-scale local entrepreneurs with the potential not only for material recovery but also for the generation of local formal employment. A large part of the affected population were victims of both the disaster itself and the subsequent management by the state’s poorly equipped social and welfare structures and questionable public policies.

The post-earthquake recovery process highlighted the weaknesses in public management, lack of appropriate regulation, poor vision for the implementation of circular economy processes, inability to take care of certain vulnerable sections of the population, and acceptance of inherent environmental damage. This should serve as an experience for authorities in other locations—not just Ecuador—to be prepared to deal successfully with natural disasters that affect the urban infrastructure. Learning from the experience of other countries is an indispensable strategy to improve and strengthen public management and land planning processes in order to reduce the impacts and consequences of crises. Such knowledge gives affected populations a greater capacity for maneuvering and the opportunity for improvement, both critical to achieving greater resilience of cities.

## Figures and Tables

**Figure 1 ijerph-18-03494-f001:**
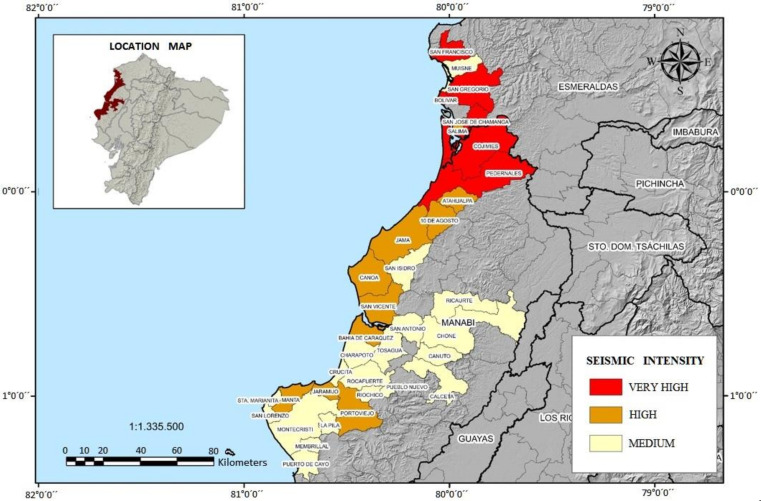
Seismic intensity map by parish. Source: Department of Risk Management [21].

**Table 1 ijerph-18-03494-t001:** Processes.

	Pedernales ^1^	Portoviejo ^2^	Manta ^3^
Process 1	- Evaluation of affected infrastructure and elaboration of the intervention schedule, prioritizing affected areas.	- Evaluation of the affected infrastructure to determine its suitability for restoration or waste.	- Evaluation of the affected infrastructure to determine its suitability for restoration or waste.
Process 2	- Opening up of roads by removing debris. - Demolition of buildings. - Laying down of treated debris as paving on some streets lacking asphalt and in two hostels.	- Removal of the first debris to the common waste tip (not generally suitable due to its limited size and the nature of the debris). - Application of the Protocol for the Process of Debris Demolition and Removal.	- Selection and repurposing of the recreational site “La Poza” on the esplanade (the closest to the affected area), for removal of debris. - Authorization of the deployment of approximately 400 salvage workers for debris classification.
Process 3	- Selection of three waste tips based on location and capacity criteria: No. 1 Cojimíes highway, No. 2 city center and No. 3 Jama highway.	- Coordination with UNDP for the selection of the site “La Solita” as an official waste tip. - Implementation of an environmental management plan to avoid significant environmental damage, which contemplated monitoring activities from the start of operations through to the end of the process.	- Selection of a ravine as a new site for the final disposal of debris, which operated only briefly due to complaints of dust and noise contamination. –Establishment of waste tips No. 1 and No. 2, the latter being a ravine located at one end of the San Juan municipal dump.
Process 4	- Authorization of the deployment of an informal group of salvage workers for the selection and separation of ferrous material from the debris.	- Authorization of the deployment of salvage workers at certain times for separation of ferrous and other recyclable materials (9 months). - UNDP backing in the form of technical recommendations and delivery of safety equipment; implementation of UNDP “Cash for Work”	- In August 2016, demolition of homes and other infrastructures was completed. - 200 people from Montecristi and Portoviejo participate in the recycling of debris. - Installation of stone crushing machinery in the waste tip.
Process 5	- Closure of the waste tip almost 3 months after its opening, with approximately 22,644 m^3^ of debris demolished and removed to its final destination. - Opening up of 23,843 m of main streets, covering an approximate area of 424 Ha. Secondary streets and rural streets not included.	- Environmental license obtained retrospectively for the operation of the tip, including its official closure. -After 10 months of operation, the separation of debris was completed at the sites, leaving only concrete materials to continue to arrive at La Solita for final disposal.	- Based on daily records (without accounting for early morning checks), it is estimated that 8 to 12 million cubic meters of material were buried in the tips. Recycled materials were used to pave secondary roads.

Source: compiled by the author from fieldwork during various periods: ^1^ July and August 2017, (C. Corral, personal interview, 14–16 July 2017); ^2^ July and August 2017, (M. Estévez, personal interview, 18–19 July 2017); ^3^ July and August 2017, (W. Navarro, personal interview, 20–22 July 2017).

**Table 2 ijerph-18-03494-t002:** Resources.

Recources	Impact
Environmental	Land resources: degradation of the natural landscape due to the loss of plant cover, decrease of natural areas, decrease of biodiversity, destabilization of land. Increased pressure and exploitation of mines and quarries for new constructions due to the non-use of recycled material (indirect impact). Water resources: Changes to natural drainage systems, pollution by untreated chemicals and bacteria, possible contamination of nearby groundwater. Air resources: Atmospheric emissions from debris handling and by loading and unloading material. Noise pollution: Uncontrolled noise from the operation of heavy machinery. Transport of heavy-duty vehicles and operation of crushers that break down the materials. Increased number of zones for final disposal of debris (indirect impact).
Visual	Deterioration of the landscape and of the natural environment: replacement of natural greenery with a bare landscape of gray hues.
Social	Impacts on the health of salvage workers: lung problems from dust absorption, possible cuts when collecting recyclable materials, ergonomic problems from loading material. Low level of acceptance of disposal work by surrounding urban areas Increasing poverty levels in sectors surrounding the waste tips.
Economic	Economic loss from wastage of the material. Loss of new sources of employment in the development of debris exploitation work. Proliferation of informal and high-risk work. Increased operating costs from heavy machinery used for disposal.

Source: compiled by the author from various interviews.

**Table 3 ijerph-18-03494-t003:** Composition and percentages.

Category	Material	% In City Studies	
		Madrid	Barranquilla
Usable	Brick, tiles and other ceramics	54	13.67
	Concrete and cement	12	18.16
	Stone	5	-
	Aggregates	4	24.11
	Wood	4	10.93
	Glass	0.5	1.52
	Plastics	1.5	3.91
	Metals	2.5	5.76
	Paper	0.3	1.77
	Cardboard	*-*	5.23
	Organic waste	*-*	3.47
	Total	83.8	88.53
Non-usables and others (not defined)	Asphalt	5	0.09
	Plaster	0.2	2.27
	Styrofoam	*-*	6.29
	Asbestos	*-*	0.96
	Rubber	*-*	0.15
	Textiles	*-*	0.1
	Garbage	7	-
	Others	4	1.63
	Total	16.2	11.49

Source: compiled by the author from Gutiérrez et al., 2014 [47], Llatas, 2013 [48], and Pacheco et al., 2011 [49] (2011). Note: the construction of this table required the grouping of common materials that were separated in the original research.

## Data Availability

Not applicable.

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
