# Peer review of "Social Responsibility, Sustainability, and Public Policy: The Lessons of Debris Management after the ManabĂ Earthquake in Ecuador"

_ijerph, 2021, doi:10.3390/ijerph18073494_

Round 1
Reviewer 1 Report
This paper presents a sound review of the issues related to debris management after disaster.
Unfortunately, the methodology for how the findings were collected is not clear in p 6 of 16 (debris). The paper mentioned that there was fieldwork analysis of the institutional strategies, however, no no analysis of the strategies was actually presented, just 'results' .
The paper should include a stronger discussion of how those findings in particular were developed. These findings are simply presented, and discussing their basis would strengthen the paper.
Further, more work comparing the case studies that are well presented in the beginning to the Equadorian experience (even if a city was eliminated) would help to strengthen the paper's findings. This paper has the makings of a seminal work if these issues are addressed.
Minor findings: in the abstract it actually should be social responsibility in the distorted processes not social (ir-)responsibility in the distorted processes. and, my copy has a strange green highlight on reference #8? it might be an artifact of my download, but that should be checked as well.
Author Response
Dear professor and colleague,
First of all, I wish to thank you for the thoroughness, kindness and courtesy you have shown in your review. I fully understand and value the work of a reviewer, which in many cases receives no other reward than the professional satisfaction of carrying out a vital task and the respect of fellow academics who, after months (or years) of work on a project, submit their papers for evaluation. As the lead author of this manuscript, I too review a number of articles throughout the year.
In this case, your indications and advice seem entirely constructive, with the clear purpose of improving the study. Understandably, we have had to balance your comments with the recommendations of the other reviewer in order to avoid contradictions in the text. I enjoy the challenge of incorporating corrections and suggestions and the process, of course, helps all writers to grow, learn and avoid future mistakes. This short period that I have been given to make all possible changes have been exhausting since making a synthesis of the informed views of two experts in a relatively short time is a mammoth task. However, even where extensive sections have needed to be altered in accordance with your review, I have diligently attempted to respond to your recommendations.
It is true that it is a very broad-ranging article but I feel it makes a valid contribution because of its theoretical basis and the depth of work invested in it. At first, I had thought merely to analyze certain environmental parameters affecting debris management, but I realized it was more worthwhile to research and present a panorama restructured according to all the suggested revisions.
The work, both the original submission and the altered version, has been very complex to carry out. The breadth of readings and the depth of analysis required were considerable, but the writing-up was especially demanding in order to make the many different aspects coherent and attractive for the reader. The phenomenon of earthquakes is extremely varied and complex, adding to the difficulty we faced. The overall purpose was to present a general overview of sustainability in Ecuador after the earthquake (which I believe we have achieved through the wide range of techniques and resources offered by the social sciences, notwithstanding the logical shortcomings of such an extensive and complex work).
The processes of responsibility, sustainability and public policy are very broad and the normative aspects of the study (in all their dimensions) are, I think, essential if we are to appreciate the scale of the problem and its implications following the earthquake. Consequently, we wish to fully address much of what you have suggested in the revised paper, that is, to analyze in greater depth those articles related to environmental aspects, as well as the non-regulatory processes (mainly at the corporate level) that can improve these standards (recycling).
Below are the specific aspects we have addressed:
- Some of the results have been reformulated.
- Specific research questions have been clarified.
-“My copy has a strange green highlight on reference #8?”—resolved
-“In the abstract it actually should be social responsibility in the distorted processes not social (ir-) responsibility in the distorted processes.”—resolved
- The methodology is explained in more precise terms and the presentation of the findings and the strategies implemented in Ecuador have been improved.
-There is a section with case studies (bearing in mind that the text is at its word-limit) and the case of Victoria, Australia, has been included.
- The amount of debris produced by the disaster has been included, noting that “Although the exact data on the amount of debris processed is unknown, the DNPD (Department of National Planning and Development), estimated there were some 3,251,150 m3 in Manabí Province, although this is an approximate figure…” 20 lines have been added detailing the data pertaining to the disaster in addition to a new table.
- Table 1 (new) shows the classification of debris from normal demolitions, which involves making provisions for planned and comprehensive waste management. The debris produced by the earthquake in the area studied are not included in this table since they are the product of a fortuitous and unpredictable event meaning that no preparation or planning for the types of waste had been carried out. The table shows that if there had been comprehensive planning and management for events of this type, such a classification could have been used indicatively.
-The state of waste management in Ecuador, both as it was previously and as it is now, has been added.
Finally, please allow me to say that I do not wish for mere acceptance for this paper: I hope to have learned from your recommendations to produce a better work that will have an impact. Your opinion matters to me and I have given much thought to each and every one of your suggestions. Thank you for your dedication, work and for the positive comments that have also given us much encouragement.
Kind regards,

Reviewer 2 Report
In this study, Table 1 describes the disaster waste response of three cities in Ecuador after the earthquake.
However, only a brief description of the seismic hazard and damage is given in 2.3. I would like to ask you to express a description of the quantitative damage and amount of disaster debris, as well as the population and normal waste management in the three cities.
Table 3 shows the composition and percentages of disaster waste. Please provide data for the three cities in Ecuador.
It is good to see that you are considering not only waste management regulations, but also land use from the perspective of disaster prevention. However, this is a qualitative study and the conclusions drawn are general: the differences in social responsibility, sustainability, and public policy in Pedernales, Portoviejo, and Manta, as well as the differences in the characteristics of the three cities, are not clear. I would like to ask you to discuss the differences in social responsibility, sustainability, and public policy in Pedernales, Portoviejo, and Manta, as well as the differences that are common to other cities in Ecuador and those that need Tobe adapted according to the characteristics of the community, and the differences between Ecuador and other countries.
Author Response

(The authors gave the same response as above.)

Round 2
Reviewer 1 Report
much improved!